# How few annotations are needed for segmentation using a multi-planar U-Net?

**William Michael Laprade**[1,2]                    WL@DI.KU.DK
**Mathias Perslev**[1]                             MAP@DI.KU.DK
**Jon Sporring**[1,2]                              SPORRING@DI.KU.DK

[1] *Department of Computer Science, University of Copenhagen (DIKU), Denmark*

[2] *Center for Quantification of Image Data from MAX IV (QIM),*
  *Technical University of Denmark (DTU), Denmark*

## Abstract

U-Net architectures are an extremely powerful tool for segmenting 3D volumes, and the recently proposed multi-planar U-Net has reduced the computational requirement for using the U-Net architecture on three-dimensional isotropic data to a subset of two-dimensional planes. Despite this considerable reduction in model-parameters and training data needed, providing the required manually annotated data can still be a daunting task. In this article, we investigate the multi-planar U-Net's ability to learn three-dimensional structures in isotropic data from sparsely annotated training samples. Technically, we pick random training planes intersecting the three-dimensional image and sparsely annotate the pixels along random lines in each of these planes. We present our empirical findings on a public domain, electron microscopy data set, which has been fully annotated by an expert, and surprisingly we find that the multi-planar U-Net with our random annotation strategy on average requires less than 30% of the annotations. Sometimes less is more!

**Keywords:** 3D imaging, segmentation, deep learning, U-Net, sparse annotations

## 1. Introduction

Deep learning methods for the segmentation of 3D image data typically require a lot of manually labeled data for training, and manual labeling is a very time-consuming process. Often this is also an inefficient process since similar structures in the images may be labeled repeatedly, even if the model could learn from fewer samples. In this paper, we investigate how well a 2D U-Net segmentation model can learn to segment 3D images from only sparsely annotated label maps.

Over the years the U-Net architecture has been altered to work on a variety of 3D imaging tasks through a number of different modifications. These range from simply adapting the architecture to use 3D convolutions (Özgün Çiçek et al., 2016) to a more complex multi-planar approach that attempts to learn the rotational properties of the data by slicing through random planes in the volume (Perslev et al., 2019). In all cases, we continue to run into the limitations that arise due to a lack of annotated data. It is both time-consuming and expensive to have experts manually annotate large 3D volumes to be able to train effective segmentation models.

There have been various other techniques to efficiently train CNN segmentation models from limited and/or sparse data. These include defining loss balancing methods to weigh

the contributions from each class equally as in (Bokhorst et al., 2019), a more complete loss function overhaul that facilitates learning from sparse data as in (Kervadec et al., 2019), or learning from bounding box annotations as in (Rajchl et al., 2017). The common aspect of many of these methods is that they attempt to extract that extra bit of performance from the data during the optimization of the model (Bokhorst et al., 2019; Kervadec et al., 2019). Here, we investigate a different approach that involves improving the dataset itself via a method more similar to data augmentation.

Aggressive data augmentation is one of the ways we can attempt to counteract the problems of limited data. The common structural data augmentations (translations, rotations, flips, shear, scaling, elastic deformations) attempt to capture the structural variability in the data while visual data augmentations (brightness, contrast, color shift, gamma adjustments, noise, Gaussian blur) attempt to capture the variability in lighting and color as well as possible imaging artifacts in the data. Elastic deformations in particular (Simard et al., 2003) have been shown to be useful in medical imaging due to the inherent deformability of biological structures (Ronneberger et al., 2015).

Structural augmentation is an attempt to estimate the source distribution of image patches, however, it is often unknown to what extent, the imposed transformations reflect the true distribution of random image-patches from the data source. The multi-planar U-Net (Perslev et al., 2019) makes use of the computational efficiency of the 2D U-Net and includes off-plane training samples, thus better estimating the true underlying distribution of image-patches in isotropic data. We argue that multi-planar sampling may maximize the use of the available information in the ground truth images by facilitating a stronger learning signal.

While typical approaches include fitting a 3D segmentation model to the 3D data directly, or a 2D model to 2D image-slices along a single axis, such approaches use each – sparsely available – label only once (i.e., with exactly one image input) in each epoch. Multi-planar sampling allows each label to appear in a large number of unique, yet biologically plausible, input image-slices. As many 2D image features are invariant to the rotation of the 3D image volume (e.g., low-level features such as edges and textures), multi-planar sampling allows the learning of such features from an expanded training dataset. For some datasets, such as the biological tissues imaged using electron microscopy that we consider here, there are no intrinsic orientations of the imaged objects within the volume. In such cases, most learned features will apply equally well to any 2D image-slice independent of its orientation within the volume.

In this paper, we investigate the multi-planar U-Net's (Perslev et al., 2019) ability to learn from sparse annotations. Our paper is organized as follows: First, we motivate and formalize our method and the sampling technique. Then we present an empirical investigation on the relation between the number of annotated pixels and the Dice (Dice, 1945; Sørensen, 1948) score on a public domain, expert-annotated electron microscopy 3-dimensional dataset. Finally, we give our conclusions.

## 2. Methods

Multi-planar training with sparse labels is an algorithm for learning a 3D image segmentation model from a training dataset with sparse ground truth segmentation maps. Our

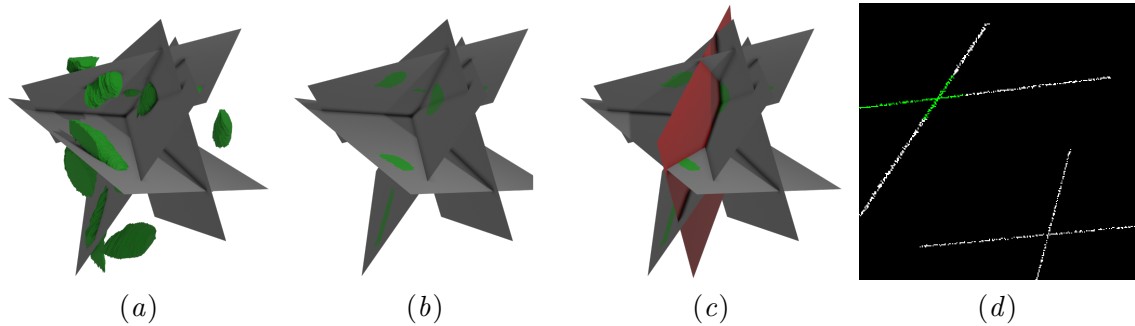

$(a)$ $(b)$ $(c)$ $(d)$

Figure 1: (a) Mitochondria (green) and 4 annotation-planes, (b) sparsely-labeled volume used for training-plane generation, (c) an example training-plane (red) that slices through the sparsely-labeled volume, (d) the weight map for a possible training-plane to define which pixels contribute to the loss computations.

method is based on multi-planar data sampling as described in (Perslev et al., 2019), which expands the training dataset by isotropic sampling of the image and label space on 2D planes of random orientation. Specifically, we fully annotate a small set of randomly selected annotation-planes from a 3D image, then we choose another set of random training-planes that are intersected with the annotation-planes. Finally, we fit a 2D fully convolutional neural network (FCN) inspired by the popular U-Net (Ronneberger et al., 2015) architecture to the training-planes only considering the annotations in the intersecting lines. This is illustrated in Figure 1. In the following sections, we will discuss the sampling strategies in detail.

## 2.1. Random annotation-planes

Given a dataset $D = \{(I_1, L_1), ..., (I_n, L_n)\}$ where $I_i \in \mathbb{R}^{w \times h \times d \times C}$ are 3D image volumes with $C$ channels and $L_i \in \mathbb{N}_0^{w \times h \times d}$ are corresponding label maps, we wish to learn a function $f : I \to \hat{L}$ where $\hat{L} \approx L$ for a new image-label map pair $(I, L)$ drawn from the same (stable) distribution that generated $D$. In practice, the label map $L_i$ is generated by a human annotator in an often time-consuming effort. Our method aims to learn an alternative, but similarly performing function, $f^*$, from an only partially annotated dataset $D^*$. Specifically, each label volume $L_i^* \in D^*$ has a defined label value at voxel $\mathbf{x} \in \mathbb{R}^{w \times h \times d}$, only if $\mathbf{x}$ is an element of a pre-determined set of indices $A_i$ where $|A_i| < w \cdot h \cdot d$.

We create each set $A_i$ by selecting $n_{\text{annot}}$ numbers of 2D planes with random orientation and location (referred to as *annotation-planes*), and use these to sample the image volume $I_i$. The selected annotation-planes are then fully annotated by a human expert and in union define the available label map. Because each annotation-plane spans volume $I_i$ at a random angle and offset from the image center, each label volume $L_i^*$ may be labeled to variable degrees. For instance, the selected annotation-planes may span beyond the domain of the image volume, or multiple planes may overlap causing some voxels to be annotated multiple times.

## 2.2. Random training-planes

The decisive feature of our method is the multi-planar sampling of the sparse datasets $D^*$. From across all images and label maps $(I_i, L_i^*) \in D^*$ we select $n_{\text{train}}$ number of planes with random orientation and location (referred to as *training-planes*), and use these to sample the image and label-map $(I_{i,j}, L_{i,j}^*)$ where $j \in [1, 2, ..., n_{\text{train}}]$. We used trilinear interpolation and nearest neighbour interpolation for sampling the image and label maps, respectively. An example of a training-plane is displayed in Figure 2.

## 2.3. Sampling heuristics

For any model to learn a segmentation task, the training set must contain examples of all classes. In our setup, we generate the annotation- and training-planes according to the following heuristics:

- Annotation-planes: Randomly chosen uniformly from the set of planes that contain at least one positive and one negative label. This ensures the model will learn both classes.

- Training-planes: Randomly chosen uniformly from the set of possible planes that contain at least one voxel from an annotation-plane. This prevents zero division in the loss computation when $N = 0$ (see Equation 1). These are sampled with a probability of 0.5 that they will contain at least one positive label to ensure there is some information regarding the positive class in the set of $n_{\text{train}}$ samples.

## 2.4. Sparse, multi-planar U-Net training

We use a U-Net based architecture with noticeable modifications from the original architecture (Ronneberger et al., 2015) being the use of nearest neighbor up-sampling blocks (Odena et al., 2016) and added batch normalization (Ioffe and Szegedy, 2015) layers. Both models have 4 down-sampling and 4 up-sampling blocks. The U-Net defined here has a total of 31,044,289 trainable parameters.

The selected training-planes are input to a 2D U-Net segmentation model with the modifications defined above. Specifically, the model $f^*(I_{i,j}; \theta)$ with parameter vector $\theta$ maps an image-slice $I_{i,j}$ to a probabilistic segmentation map $\hat{L}_{i,j} \in \mathbb{R}^{256 \times 256 \times K}$ with $K$ classes (referred to as *prediction-planes*). $K = 2$ in our experiments. No information was supplied to the model regarding the orientation of each input slice. Consequently, to successfully minimize the loss function, the segmentation model must learn to segment the image volumes as seen from all orientations defined in the training-planes.

We minimize a masked cross-entropy (CE) loss function defined by,

$$E(\hat{L}_{i,j}, L_{i,j}^*) = \frac{1}{N} \sum_{\mathbf{x}} w_{\mathbf{x}} \text{CE}(\hat{L}_{i,j,\mathbf{x}}, L_{i,j,\mathbf{x}}^*), \tag{1}$$

where $\mathbf{x}$ is an index into the 2D image-slice, $w_{\mathbf{x}}$ is 1 if the voxel at index $\mathbf{x}$ is annotated (i.e., $\mathbf{x} \in A_i$) and 0 otherwise, and $N = \sum w_{\mathbf{x}}$ is the number of such annotated pixels in the given slice. We evaluated $L$ on batches of 4 slices. Data augmentations are performed on the fly and include a combination of flips, rotations, scaling, brightness, contrast, gamma

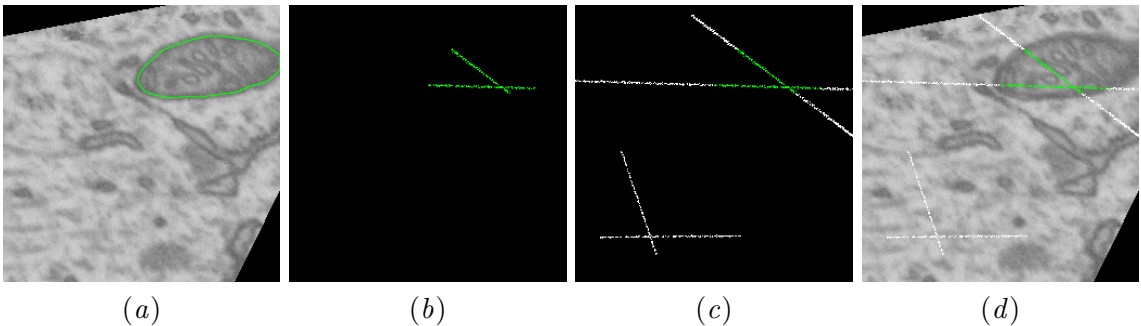

$(a)$ $\qquad\qquad$ $(b)$ $\qquad\qquad$ $(c)$ $\qquad\qquad$ $(d)$

Figure 2: An example of a training-plane taken from the electron microscopy mitochondria dataset. (a) Randomly oriented slice through the sparsely annotated volume, (b) the corresponding label map, (c) the corresponding binary weight map, (d) label and weight maps overlayed on the image-slice.

adjustments, and random noise. We used the Adam optimizer (Kingma and Ba, 2015) with a learning rate of $\eta = 10^{-5}$ and default $\beta_1 = 0.9$, $\beta_2 = 0.999$ and $\epsilon = 10^{-7}$ parameters. We monitored the performance of the model on a held-out validation set of $n_{\text{annot}}/2$ image-slices. Optimization continued for 22400 gradient updates. The best observed model (as per validation performance) was selected for further analysis on a held-out test set.

At test-time, the learned invariance is exploited further by segmenting the target volumes multiple times. Specifically, the model is used to predict along 3 randomly chosen views, $V = \{v_1, v_2, v_3\}$ with $v \in \mathbb{R}^3$ and $||v|| = 1$, establishing a set of 3 proposal segmentation volumes $\mathbf{P} = \{\, P_v \in \mathbb{R}^{w \times h \times d \times K} \mid v \in V \,\}$. Each $P_v$ is re-aligned with the voxel grid and the single volume $P = \arg\max_K(\text{mean}(\mathbf{P}))$ is considered the final segmentation.

## 3. Datasets

For empirical evaluation of the network's ability to learn from a varying number of annotations, we consider the task of mitochondrial binary segmentation in electron microscopy images of a $5 \times 5 \times 5\,\mu m$ CA1 hippocampus region of a rodent brain (Lucchi et al., 2013). The dataset consists of two annotated $165 \times 768 \times 1024$ sub-volumes with a voxel resolution of approximately $5 \times 5 \times 5$ nm. The two volumes define the training and testing sets of our experiments, respectively. Given that we have ground truth annotations available for the full volumes, we simulate the process of creating sparse dataset $D_1^*$ (i.e., labeling only the voxels in $A_i$) by taking values along these randomly oriented annotation-planes from the already annotated volumes and ignoring all other annotations. Finally, we evaluate our method on a separate cardiac dataset (Simpson et al., 2019) consisting of 20 mono-modal MRIs of dimension $320 \times 320 \times z$, where z varies between 90 and 130 depending on the scan. The cardiac dataset tests if the network can learn to segment objects with high shape variation across views.

## 4. Experiments and Results

We tested the performance of models obtained from our proposed method as a function of two hyperparameters:

- Degree of label sparsity: We fitted and evaluated models trained on dataset $D^*$ with each label map $L_i^*$ defined in $A_i$ spanning $n_{\text{annot}} = \{4, 8, 12, 16\}$ random image-slices.

- Number of sampled image planes: From each sparsely annotated dataset, we generated $n_{\text{train}} \in \{0, 128, 256, 384, 512\}$ random 2D image-slices using the multi-planar sampling algorithm. $n_{\text{train}} = 0$ is a notation we use for the base case, where the annotation- and training-planes coincide, such that we do not make use of the off-plane sampling but train on the fully annotated annotation-planes.

We tested all combinations of $n_{\text{annot}}$ and $n_{\text{train}}$ for a total of 20 experimental setups. When the number of sampled training-planes $n_{\text{train}} = 0$ the model is fit to only the $n_{\text{annot}}$ fully annotated planes. In all other cases, each model observes *only* the sampled sparse training-planes.

**Mitochondrial segmentation:** We evaluated the performance of each experimental configuration on the held-out test set containing four $165 \times 165 \times 165$ volumes. The training volume is split into 4 sub-volumes of dimensions 165 x 448 x 448. We did 12 experimental runs using train-validation pairs from these 4 volumes. In each repetition, the sparse label volumes $L_i^*$ were randomly re-created by selecting new random annotation-planes from each image volume $x_i$. A new training dataset was created in each repetition by sampling a new set of training-planes. The parameter vector $\theta$ was randomly initialized before training in each repetition. We report the mean and standard deviation F1/Dice (Dice, 1945; Sørensen, 1948) scores and the mean and standard deviation of the number of annotated pixels across the 12 repetitions. For further validation, additional metrics (sensitivity, specificity, average surface distance) as well as examples of predicted masks are included in Appendices A and D, respectively.

As was to be expected, the performance for the mitochondria segmentation task improves as both the number of initial annotation planes and the number of generated training planes increases, see Figure 3.

As a peculiarity, we note that models trained with $n_{\text{annot}} = 4$ and $n_{\text{train}} = 0$ have the highest variability (standard deviation of 0.2280 and median absolute deviation of 0.1106) and lowest scores (mean of 0.6493 and median of 0.6979). This suggests a difficulty during optimization and is the result of some of the experimental runs overfitting to the 4 samples in the training set, thus performing poorly on the held-out test set.

Surprisingly, we see that we can achieve nearly the same performance with $n_{\text{annot}} = 4$ and $n_{\text{train}} = 512$ as we can with $n_{\text{annot}} = 16$ and $n_{\text{train}} = 0$ (0.8974 and 0.8985 respectively). Consequently, this means that with this dataset we can achieve roughly the performance of 16 manually annotated planes with only 25% of the manual segmentation effort. Interestingly, models trained with $n_{\text{annot}} = 4$ and $n_{\text{train}} = 512$ have, on average, more than 70% fewer annotated pixels in the dataset than those trained with $n_{\text{annot}} = 16$ and $n_{\text{train}} = 0$ yet the model still performs just as well. Impressively, the highest Dice score achieved, 0.9245, (when $n_{\text{annot}} = 16$ and $n_{\text{train}} = 512$) is comparable to the score of 0.9288 achieved with full supervision in (Lucchi et al., 2013).

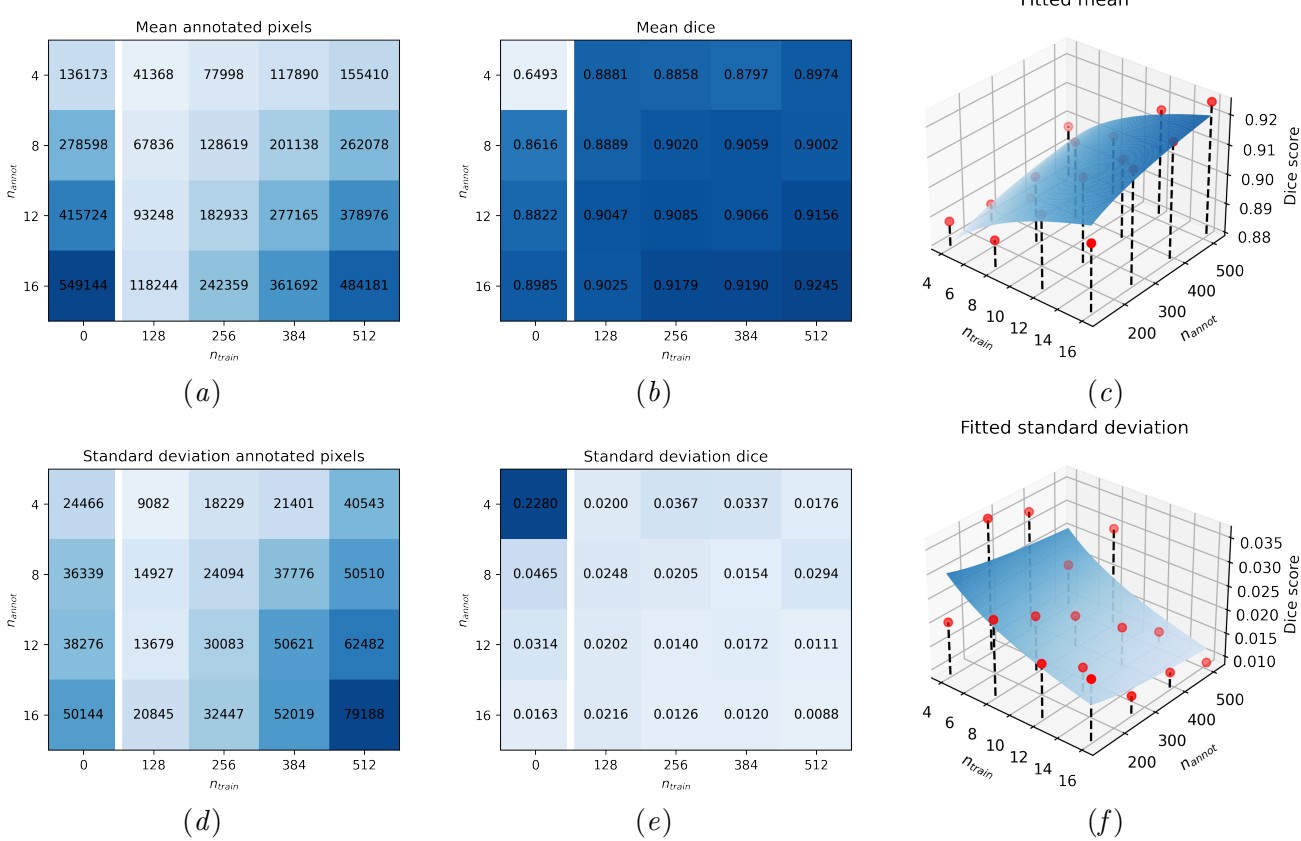

Figure 3: (a) Mean number of annotated pixels in the set used to train each model, (b) Mean Dice scores on the held-out test set, (c) Best fit plane through mean Dice scores using log of $n_\text{annot}$ and $n_\text{train}$, (d) Standard deviation of the number of pixels used to train each model, (e) Standard deviation of Dices scores on the held-out test set, (f) Best fit plane through the standard deviations of Dice scores using log of $n_\text{annot}$ and $n_\text{train}$.

Finally, we fitted functions to the mean and standard deviation of the Dice scores as a function of $n_\text{annot}$ and $n_\text{train}$:

$$f_\text{mean}(n_\text{annot}, n_\text{train}) = 0.02025 \cdot \log n_\text{annot} + 0.00848 \cdot \log n_\text{train} + 0.81093 \tag{2}$$

$$f_\text{std}(n_\text{annot}, n_\text{train}) = -0.00999 \cdot \log n_\text{annot} + -0.00319 \cdot \log n_\text{train} + 0.05952 \tag{3}$$

These functions are visualized in the last row of Figure 3 on a non-logarithmic scale. Although based on only a few samples, the increase in mean Dice score and decrease in standard deviation by $n_\text{annot}$ and $n_\text{train}$ seems to follow a simple law, which supports the view, that the more unique planar views that we give to the model, the more likely we will arrive at a well-converged model. It is of course expected that beyond our tested range ($n_\text{annot} > 16$ and/or $n_\text{train} > 512$) we will begin to see significant diminishing returns.

**Cardiac segmentation:** We performed a similar set of experiments on the caridac dataset. Out of the 20 available MRIs four were set aside and used as the held-out test set. The remaining 16 volumes were divided into 12 training volumes and 4 validation volumes. This

was repeated 12 times with different train-validation splits. The results are included in Appendix B. Overall, the results follow the pattern observed on the mitochondria dataset with increasing performance as a function of both $n_{\text{train}}$ and $n_{\text{annot}}$. With $n_{\text{annot}} = 16$ and $n_{\text{train}} = 512$ the model achieves a mean dice score of 0.83. For comparision, a fully supervised multi-planar model achieves approximately 0.89 (Perslev et al., 2019) although on a separate, non-public test-set.

**3D U-Net:** For comparison, we also trained 3D U-Net models with equivalent sparsity on the mitocondrial dataset. Results and additional training details are included in Appendix C. Evaluation numbers result from a single, non-repeated evaluations of each experimental setting. Based on these preliminary results, the 3D U-Net also performs well on the highly sparse datasets with maximum observed dice scores matching those of the multi planar model. However, we observe larger variability and lower minimum dice scores compared to the multi-planar U-Net. The training time for the 3D model takes approximately 2 hours and 45 minutes, while the multi-planar version takes 1 hour and 5 minutes.

## 5. Discussion

Segmenting data in 3D is usually the first step in analyzing 3D medical data. As such, we must do this in a way that is both time-efficient for the doctor/researcher as well as accurate enough for their needs. The study and method described in this paper are important to give us an understanding of how we can improve segmentation performance without any additional manual effort.

With the multi-planar U-Net (Perslev et al., 2019) we can achieve a good segmentation of a 3D volume via a 2D U-Net from only sparsely annotated samples. The multi-planar U-Net has two important advantages over traditional 2D U-Nets applied to 3D data:

- Not only can we learn in-plane rotational invariance via rotational data augmentations, but out-of-plane rotational invariance via planar sampling is also learned. As a result, the multi-planar method is better than a standard 2D U-Net at learning 3D structures in the data regardless of their local orientations.

- Sparse sampling also increases dataset variability while reducing the number of repeated annotations, thus preventing overfitting to these repeated structures.

It is evident from the results above, that improving the segmentation accuracy is as simple as creating a more varied, less repetitive dataset. One interesting observation is that even though the tests with up to $n_{\text{train}} = 512$ perform better, they have, on average, fewer annotated pixels than the $n_{\text{annot}}$ samples used to generate them (except when $n_{\text{annot}} = 4$ and $n_{\text{train}} = 512$). This suggests that it is less important to have a lot of total data (due to repeated annotations of similar structures), and more important to have a lot of unique data (with a large variation in annotated structures). Intrinsically, this makes sense because as we increase $n_{\text{train}}$ we increase the number of unique views, thus generating a lot of 2D structural variation in the dataset.

In summary, we provide evidence that with a small initial set of samples we can increase the performance of a model by generating a unique dataset with fewer annotated pixels per sample, but more variation in viewing angles. Thus, sometimes less is more.

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

## Appendix A.  Additional metrics

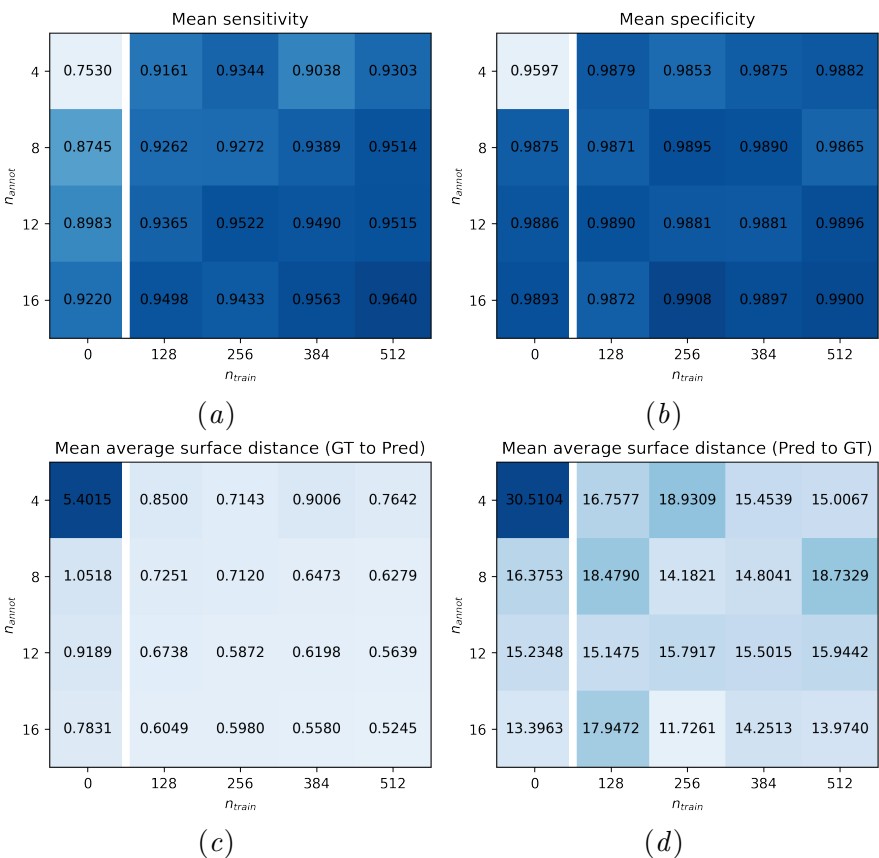

Figure 4: Additional metrics on the mitochondria dataset (a) Mean sensitivity, (b) Mean specificity, (c) Mean average surface distance from ground truth surface to predicted surface, (d) Mean average surface distance from prediction surface to ground truth surface

## Appendix B. Cardiac dataset

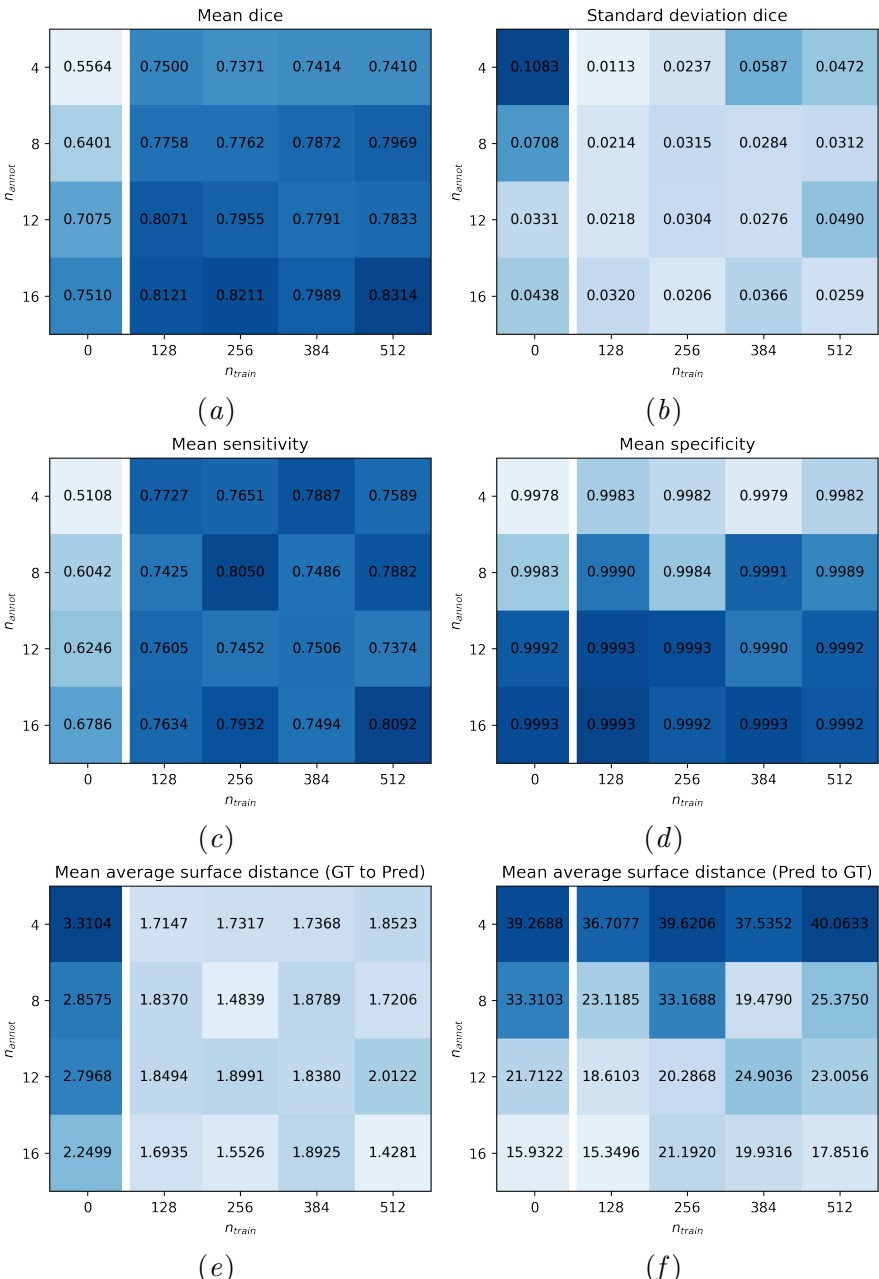

Figure 5: Results on cardiac dataset (a) Mean Dice scores (b) Standard deviation of Dices scores (c) Mean sensitivity, (d) Mean specificity, (e) Mean average surface distance from ground truth surface to predicted surface, (f) Mean average surface distance from prediction surface to ground truth surface

## Appendix C. 3D U-Net results

For additional validation, we compared our multi-planar method to a 3D U-Net trained on the same levels of sparsity using the mitochondria dataset. The 3D model is identical to the one outlined in the 3D U-Net paper (Özgün Çiçek et al., 2016) with approximately 19 million parameters. Models were trained for 22400 batch updates with a batch size of 1 and input size of $64 \times 64 \times 64$. The input size and batch size were chosen to be consistent with the multi-planar versions such that both models see 262144 pixels during each batch update. The best model is saved based on the validation loss. Data augmentations are identical to the multi-planar version. The models were trained on the same data seen by the multi-planar version (i.e. the annotated voxels are the ones that appear in the intersection of the $n_{annot}$ voxels and the $n_{train}$ voxels when $n_{train} > 0$ and the $n_{annot}$ voxels alone when $n_{train} = 0$). Prediction is done in patches aligned with the voxel grid.

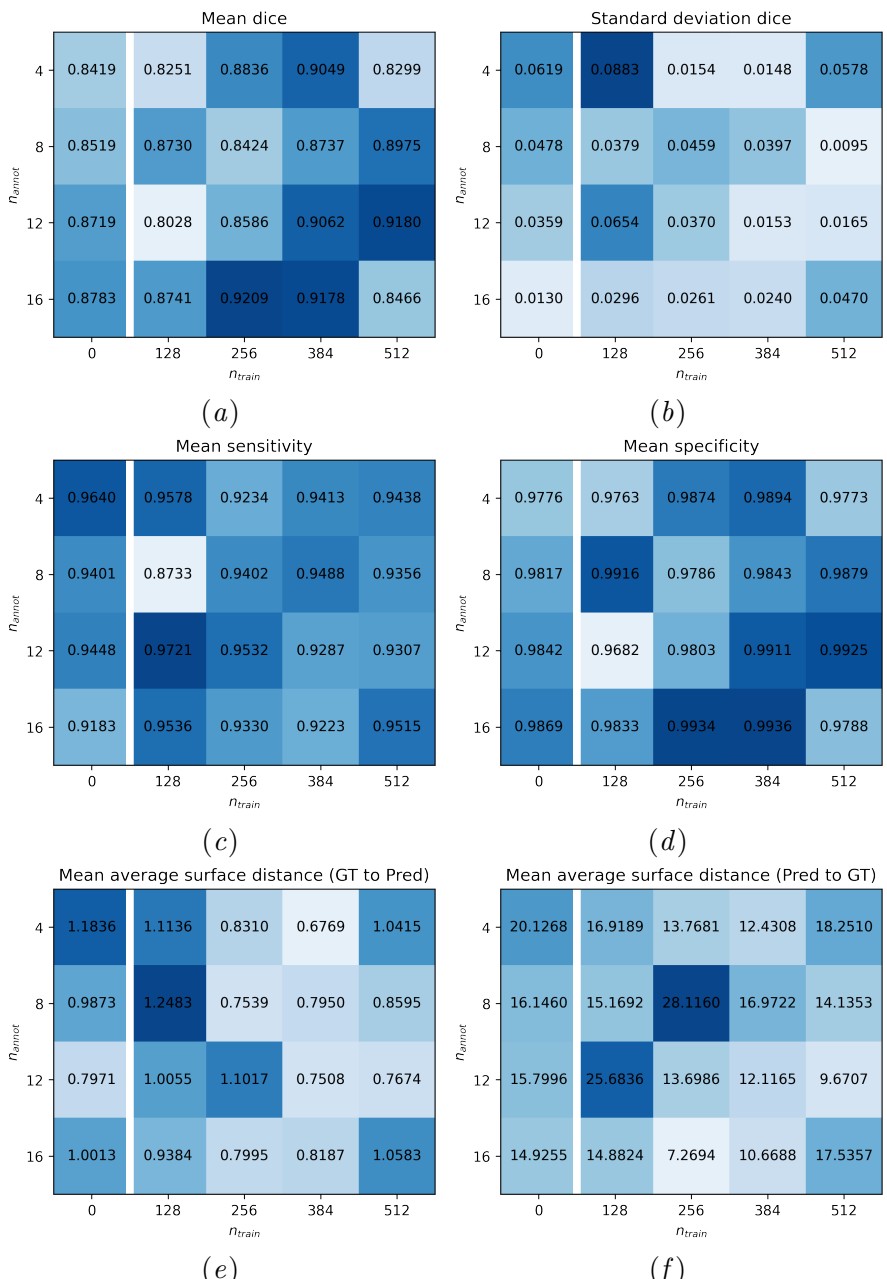

Figure 6: 3D U-Net results on mitochondria dataset (a) Mean Dice scores, (b) Standard deviation of Dices scores, (c) Mean sensitivity, (d) Mean specificity, (e) Mean average surface distance from ground truth surface to predicted surface, (f) Mean average surface distance from prediction surface to ground truth surface

## Appendix D.  Example predictions

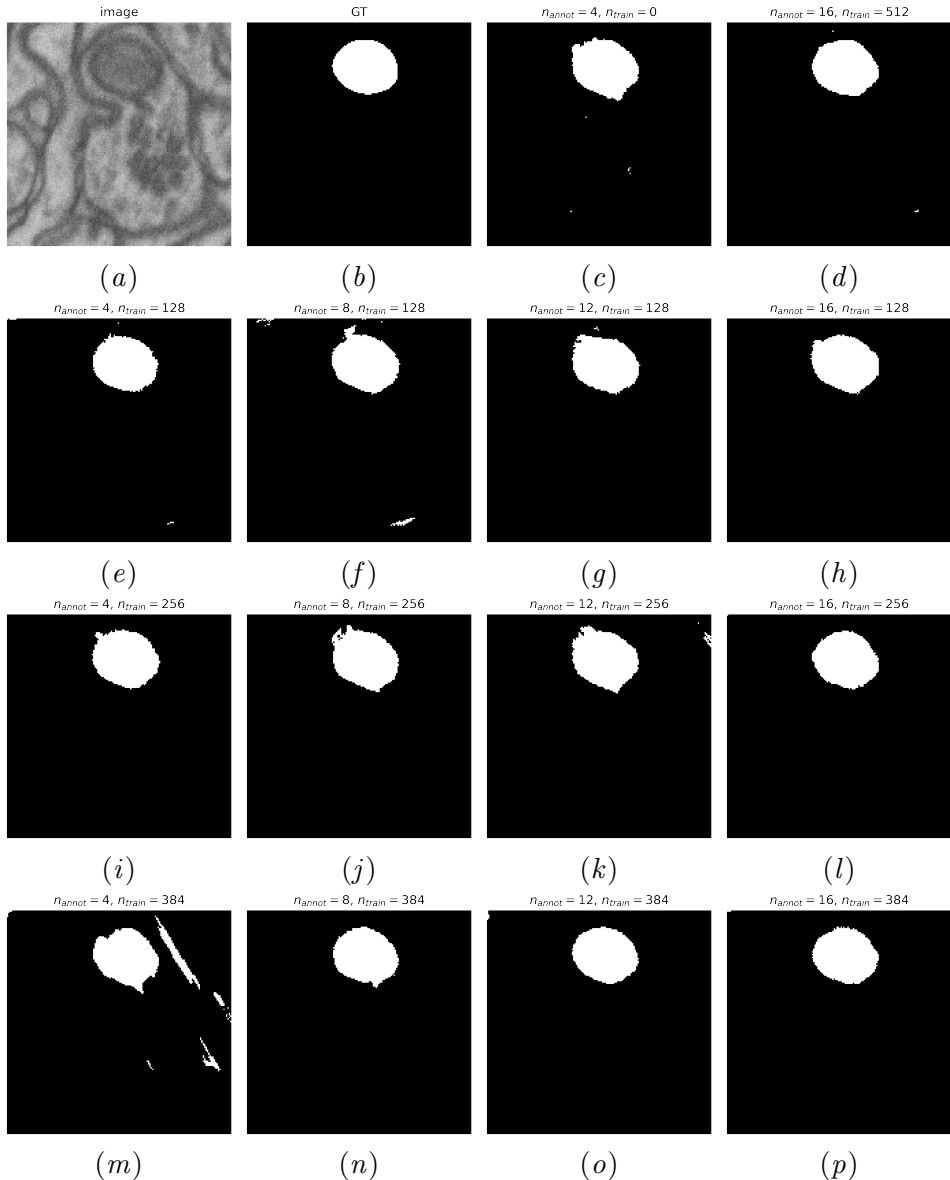

Figure 7: Segmentation results across a number of different $n_{annot}$ and $n_{train}$ combinations on the mitochondrial dataset using the multi-planar model.

