# OpenReview forum: "How few annotations are needed for segmentation using a multi-planar U-Net?"
_MIDL.io/2021/Conference — Submitted to MIDL 2021_

### Official Review · AnonReviewer2 · 2021-03-08

**Confidence:** 4
**Preliminary Rating:** 3
**Final Rating:** 3

**Summary:**

In this paper, the authors investigate how much labeled data is needed to effectively train a multi-planar U-Net, which is a method published in MICCAI in 2019. The method extracts randomly oriented planes through the 3D data, annotates these, and in combination with augmentation uses that to train the multiplanar 2D U-Net. In the experiments of the paper, the number of planes that are annotated, and the number of augmentations is varied to see how that affects the final results. The paper concludes that more variation in the annotated planes is beneficial.

**Strengths:**

The paper presents the method clearly, provides a good design of the experiments, and has a thorough analysis of the effect when the different parameters of the method are varied. The paper explains the MultiPlanar 2D U-Net and how it performed on previous benchmarks well. The paper is well written.

**Weaknesses:**

- I am missing a good comparison with the 3D U-Net paper by Cicek et al. That paper also learned from sparse samples, so it is in fact very comparable. In that paper, the number of sparsely annotated samples needed is not investigated, so that's where this paper adds value. However, I would have liked to say a better comparison and discussion on how this paper, and for example the loss used in this paper, compares to that paper.
- The results show the improvements when the number of sparsely annotated samples increases, but does not show what the performance of the baseline system would be (just training the Multi-Planar UNet on the full data). Also, it would be great to see how the 3D U-Net would do on this task, and the nnUnet. Then, it is more easy to compare these results with current SOTA.
- The paper has little technical novelty, but does a good job in investigating the optimal number of data needed to train a multiplaner U-Net for this task.

**Deanonymize Review:**

no

**Detailed Comments:**

- For sampling the annotation planes, the planes are randomly chosen uniformly from the set of planes that contain
at least one positive and one negative label. For a completely new dataset, you would not know whether a certain plane would contain at least one positive and one negative label. So, this would have to be done by the annotators? They would only annotate planes on which a positive and negative label would be present, and would exclude others?
- When n-train is set higher than zero, each model observes only the sampled sparse training planes. Why is this? I would expect that n-train is 128 would also use the original annotated training planes because that's what you would do in practice, because the additional planes are basically augmentation samples?
- I am not surprised that numbers lower than 128 does not work for the training planes. If you look at Figure 2C, there are only very few annotated samples for some of these planes.
- The planes do not necessarily have to go through the center of the image, but how far from the center can they be? No limitation so could be completely at the edge of the 3D volume?

**Final Rating Justification:**

I am satisfied with how the authors addressed my concerns. I still have the opinion that the paper presents little technical novelty, but it now has a better discussion with respect to other recent SOTA work in this area, such as 3D U-Net, and is therefore of interest to MIDL.

**Justification Of The Preliminary Rating:**

The paper presents a thorough analysis, but has little technical novelty. In addition, it is lacking a performance comparison to current SOTA methods, and is missing a methodological comparison with the 3D U-Net paper.

**Paper Type:**

methodological development

**Questions To Address In The Rebuttal:**

- Comparison with 3D U-Net and nnUnet are needed.
- Please address the questions from my detailed comments section

**Special Issue:**

no

---

> ### Author Response · Authors · 2021-03-17
> **Comment for AnonReviewer2**
>
> Thank you for the feedback. We have taken it into consideration and made some necessary changes.
>
> Please read the comment to all reviewers first for some important changes, clarifications and corrections.
>
> Responses to "Weaknesses"
> - We agree that a comparison to the 3D U-Net is important and have done so to further solidify our claims. Please see the appendix and the comment to all reviewers for detailed explanation of 3D U-Net results. In general, our method performs similarly to the 3D U-Net however with a reduced training time. Although the loss function in the 3D U-Net paper (Cicek, et al) is not explicitly defined we believe that it is identical to ours based on the fact that it explicitly weights unlabeled voxels with zero in order to ignore them during loss computation.
> - We agree that a comparison to the fully supervised version of the multi-planar U-Net would be nice. Due to time constraints, we did not complete this for the mitochondria dataset. However, as seen in (Lucchi et al), they achieve a score of 0.9288 using full supervision (although not the multi-planar method) which is comparable to the 0.9245 dice score we achieve with $n_{annot} = 16$ and $n_{train}=512$. Additionally, after having performed similar experiments on an additional dataset (cardiac left atrium segmentation task from http://medicaldecathlon.com/, see comment to all reviewers and appendix for more details) we achieve a dice score of approximately 0.83 when $n_{annot}=16$, and $n_{train}=512$ while the fully supervised multi-planar method results obtained in (Perslev, et al.) are approximately 0.89.
>
> Responses to "Detailed comments"
> - In the case that the annotations are not available, we would indeed expect that the annotator should make sure to have a few samples that contain all the classes. We envision a kind of active learning system where a random slice through the volume is displayed to the annotator and they can choose to "skip" the slice if it doesn't contain both classes (at least for the first few slices to ensure all classes are present in the dataset).
> - When $n_{train} > 0$ we only view the augmented samples and disregard the annotation planes. The reason we do this is so that we can study how much information we can extract from these partially annotated samples and avoid concerns that the model might just be primarily learning the information contained in the original annotation planes (because those planes are fully annotated). You are correct, that, in a real-world use case those annotation samples would be included during training.
> - In actuality, we did not do too much experimentation with fewer than 128 training planes, so it is entirely possible that we could still achieve decent results with only 64 training planes. This is something that would be interesting to investigate further.
> - Regarding sampling limitations: We define that origin of the plane must be in the central 80\% of the volume to avoid samples with too few pixels to annotate. This means that in a 100 x 200 x 200 volume. The origin can only be in the central 80 x 160 x 160 region of the volume. This is a hyper-parameter in the sample generation phase that can be easily altered to account for larger volumes.
>
> Responses to "Questions to Address in the Rebuttal"
> - As you and other reviewers have suggested we have completed a comparison to the 3D U-Net to solidify the effectiveness of our method. In summary, the 3D U-Net achieves similar performance to our method albeit with a longer training time. See the comment to all reviewers for detailed results and the appendix of the revised paper for additional metrics.
> - See above for responses to the detailed comments section.

---

> > ### Comment · AnonReviewer2 · 2021-03-20
> > **Comment to authors**
> >
> > Thank you for the clarifications and I want to thank the authors for adding the results with 3D U-Net.

---

### Official Review · AnonReviewer1 · 2021-03-09

**Confidence:** 5
**Preliminary Rating:** 2
**Recommendation:** Poster

**Summary:**

Authors explore the ability of Multi-Planar U-Net to learn 3D structures from sparsely annotated data. Authors pick random training planes intersecting the 3D image and sparsely annoyed the pixels along random lines in each of these planes and showed that 30% of the time can be saved with this method. Experiments are b based on electron microscopy data from public resources and results are empirical.

**Strengths:**

-- sparsely annotated data in segmentation problem is a real strength because full labelled data is hard to obtained and expensive in all meanings.
-- Segmentation is an important problem, and authors focus on the algorithm, it means the algorithm can be applied to several other domains as well (different segmentation problems, detection etc)
--sampling technique that authors propose seems simple, interacted lines (randomly) chosen where positive and negative pixels available (negative is always there, in fact they only look for positive ones), and half of the pixels are randomly chosen for training.


**Weaknesses:**

---sparsely annotated data for segmentation is not a new idea,  therefore the novelty is really to apply this concept to multi-planar U-Net, which limits the contribution and scope of the idea of sparse annotation.
--Sparse annotation seems plausible however the way authors chooses sparsity does not bring any practicality; for instance, if authors focus on weakly supervised labels, that is understandable, but in this case it is not clear how labels will be obtained with this method in reality? In real settings, weak labels, noisy labels, and missing labels are common. In authors approach, it is like half of the boundaries are obtained and this has been done in two dimensions, it means structure of the segmented object is really there. Therefore, the results are not surprising.
-- study is very limited to only one data set, and its effectiveness is not known for general use.
--3D segmentation become de facto standard, not pseudo 3D....how can this method be generalized to true 3D segmentation ?


**Deanonymize Review:**

no

**Justification Of The Preliminary Rating:**

---sparsely annotated data for segmentation is not a new idea,  therefore the novelty is really to apply this concept to multi-planar U-Net, which limits the contribution and scope of the idea of sparse annotation.
-- conclusion is not complete, dice maybe not a complete metric to look, additional analysis will be necessary to see what is happening in random sampling and in which direction (or geometry that the system works best or worst)
--not clear why very particular data set is selected, generalizability of the method is questionable.

**Paper Type:**

validation/application paper

**Questions To Address In The Rebuttal:**

-- dice maybe misleading, better to give both True Positive and False Positive Volume fractions (or sensitivity and specificity)
--in what objects this method works best and in what portion this does not work good? not clear, no conclusion.
--what if the pixels are anointed in noisy manner? can authors detoriate some pixels and see how sparsity affects the results with weak labels?
-- study is very limited to only one data set, and its effectiveness is not known for general use. Do authors have tried this method in other data sets as well?
--3D segmentation become de facto standard, not pseudo 3D....how can this method be generalized to true 3D segmentation ?

**Special Issue:**

no

---

> ### Author Response · Authors · 2021-03-17
> **Comment to AnonReviewer1 - Part 1**
>
> We would like to thank the reviewer for the helpful feedback. We have taken it into consideration and made some necessary changes to the article.
>
> Please read the comment to all reviewers first for some important changes, clarifications, and corrections.
>
> Comments on "Strengths"
> - sampling technique that authors propose seems simple, interacted lines (randomly) chosen where positive and negative pixels available (negative is always there, in fact they only look for positive ones), and half of the pixels are randomly chosen for training.
>    - We would like to thank the reviewer for their detailed feedback. We noticed this comment and wondered if you may have slightly misunderstood the sampling technique. As such we will try to clarify if this is indeed the case. The statement that "half of the pixels are randomly chosen for training" is not true in our method. To attempt to clarify our process, we randomly select $n_{annot}$ annotation planes and annotate them. We now have a sparsely annotated volume with only the voxels along the annotation-planes labeled (see Figure 1b in paper). We then slice this sparsely annotated volume along new training planes. An example of a training-plane slicing through the sparsely annotated volume is in Figure 1c. This is how the lines are generated via the intersection between these annotation-planes and training-planes. Finally, the sampling heuristics are defined such that 50% of the training planes must have at least one pixel in the positive class, while the remaining 50% have no such requirement. Only the pixels along the intersection lines are used during training. Recall the terminology we defined in the paper, that training-planes are the sparsely labeled samples containing lines while the annotation-planes are fully annotated samples used to create a sparsely annotated volume.
>
> Comments on "Weaknesses"
> - Sparse annotation seems plausible however the way authors chooses sparsity does not bring any practicality; for instance, if authors focus on weakly supervised labels, that is understandable, but in this case it is not clear how labels will be obtained with this method in reality? In real settings, weak labels, noisy labels, and missing labels are common.
>    - This is a great question. We envision an active learning method where the user is given randomly oriented slices and/or grid oriented slices and asked to annotate them slowly filling regions of the volume with labels until a sufficient number of labels are obtained for accurate segmentation for a given task.
> - In authors approach, it is like half of the boundaries are obtained and this has been done in two dimensions, it means structure of the segmented object is really there. Therefore, the results are not surprising.
>    - We disagree with the statement that half the boundaries are already obtained, especially with regards to volumes with more structural variability across different viewing angles such as the cardiac dataset we looked into. The only voxels we have available in our sparse sampling are those defined in the intersection of the annotation-planes and training-planes.

---

> ### Author Response · Authors · 2021-03-17
> **Comment to AnonReviewer1 - Part 2**
>
> Comments on "Questions To Address In The Rebuttal"
> - dice maybe misleading, better to give both True Positive and False Positive Volume fractions (or sensitivity and specificity)
>    - We agree that dice may be misleading in some cases and have therefore included sensitivity and specificity as additional metrics in the appendix.
>    - Sensitivity:
> |            | n_train=0 | n_train=128 | n_train=256 | n_train=384 | n_train=512 |
> |------------|-----------|-------------|-------------|-------------|-------------|
> | n_annot=4  | 0.7530    | 0.9161      | 0.9344      | 0.9038      | 0.9303      |
> | n_annot=8  | 0.8745    | 0.9262      | 0.9272      | 0.9389      | 0.9514      |
> | n_annot=12 | 0.8983    | 0.9365      | 0.9522      | 0.9490      | 0.9515      |
> | n_annot=16 | 0.9220    | 0.9498      | 0.9433      | 0.9563      | 0.9640      |
>     - Specificity:
> |            | n_train=0 | n_train=128 | n_train=256 | n_train=384 | n_train=512 |
> |------------|-----------|-------------|-------------|-------------|-------------|
> | n_annot=4  | 0.9597    | 0.9879      | 0.9853      | 0.9875      | 0.9882      |
> | n_annot=8  | 0.9875    | 0.9871      | 0.9895      | 0.9890      | 0.9865      |
> | n_annot=12 | 0.9886    | 0.9890      | 0.9881      | 0.9881      | 0.9896      |
> | n_annot=16 | 0.9893    | 0.9872      | 0.9908      | 0.9897      | 0.9900      |
> - in what objects this method works best and in what portion this does not work good? not clear, no conclusion.
>    - We agree that this would be interesting to know, and is something we are planning on looking into in the future. However, given the limited rebuttal time, we are not able to do this analysis.
> - what if the pixels are anointed in noisy manner? can authors detoriate some pixels and see how sparsity affects the results with weak labels?
>    - This is in fact a very interesting aspect to look into however it is again something that cannot be completed given the current time constraints.
> - study is very limited to only one data set, and its effectiveness is not known for general use. Do authors have tried this method in other data sets as well?
>    - We agree that this is a very important question and is one that we have provided additional results to answer. We applied the technique on a new dataset of cardiac MRI scans from http://medicaldecathlon.com/. See the table in the comment to all reviewers for detailed results and more information in the Appendix of the revised paper. The results on this new dataset further validate our method as we see that we can improve the performance from 0.7510 to 0.8354 by simply generating more training planes (when $n_{annot} = 16$).
> - 3D segmentation become de facto standard, not pseudo 3D....how can this method be generalized to true 3D segmentation ?
>    - This question is unclear to us, and we are unsure how to answer it. However, regarding the learning of 3D structures from 2D data, we already know that 3D data can be represented as a sequence of 2D slices (see CT reconstruction or simply the fact that a volume is a series of slices), therefore it could be expected that a 2D model may begin to recognize 3D structures from 2D slices. We would like to point out that we did do additional experiments to compare to a 3D U-Net and we achieve similar results at the same degree of sparsity. See the comment to all reviewers for more specific details on the results of the 3D U-Net.

---

### Official Review · AnonReviewer6 · 2021-03-10

**Confidence:** 4
**Preliminary Rating:** 1

**Summary:**

The authors propose to sparsely annotate datasets on randomly chosen planes to reduce the annotation effort. They evaluate their method on one 3D volume and show how the parameters of the sparse label sampling influence performance. Although the paper is clear and the experiments well designed, this work brings very limited insight. The segmentation performance is not compared to full supervision or to other methods leveraging sparse annotations. In addition, the test set is too small and the claims are not backed up with statistical tests.

**Strengths:**

-Aside from the too small test set, the experiments were well designed.
-The method is original and well explained.














**Weaknesses:**

-Using this technique, the model cannot learn 3 shapes.
-The method is evaluated in a single image from a single dataset.
-No statistical test to support the claims
-We don't much how the method compares to full supervision or to other methods trained with the same amount of sparse labels. We only get to know the hyperparameters influence the method.
-To use the method in practice, the plane needs to be selected first, and a rater needs to annotate those planes subsequently. The process is not ideal: it cannot just leverage already existing sparsely labelled data.


**Deanonymize Review:**

no

**Detailed Comments:**

-The authors state " As a result, the multi-planar method is better at learning 3D structures in the data regardless of their local orientations." There is no results to support this in the paper. And intuitively it is strange to expect a 2D model to learn 3D shapes.
-The log fitting in Equation (2) and (3) seems to be a stretch. This whole section is quite an overkill to state something relatively straightforward " that the more unique planar views that we give to the model, the more likely we will arrive at a well-converged model.".


**Justification Of The Preliminary Rating:**

The results bring only very limited insight because the test set is too small, and there is no comparison with alternative methods. We don't if and how much the method actually works.












**Paper Type:**

methodological development

**Questions To Address In The Rebuttal:**

The authors should evaluate their methods in other dataset, compute statistical tests and compare to full supervision.

**Special Issue:**

no

---

> ### Author Response · Authors · 2021-03-17
> **Comment for AnonReviewer6**
>
> Thank you for the feedback. We have taken it into consideration and made some necessary changes.
>
> Please read the comment to all reviewers first for some important changes, clarifications, and corrections.
>
> Comments on "Weaknesses"
> - Using this technique, the model cannot learn 3D shapes
>      - We understand the viewpoint expressed here however we disagree with this statement. The effectiveness of this multi-planar method across a range of datasets has been shown in (Perslev et al.). We have two examples where 3D structures are captured in 2D data which may hint at the effects that may be at play. This first is in reconstruction from projections performed in tomography and the other is in representing volumes as a series of slices. In both cases, 3D data is represented as a series of 2D images, thus it is evident that 3D information can be encoded in 2D slices. So it is not unexpected that a 2D model can begin to learn to recognize 3D shapes.
> - The method is evaluated in a single image from a single dataset
>      - We understand the concern raised here and have extended the evaluation of our method with an additional dataset. Please see the comment to all reviewers for a summary and the appendix in the revised paper. In summary, the performance improvement as we increase the number of $n_{train}$ and $n_{train}$ follows a similar pattern to what we found with the mitochondria dataset, thus solidifying the validation of our method.
> - No statistical test to support the claims
>      - We agree that some statistical tests would be nice to have to back up our claims but currently do not have time to create these before the deadline.
> - We don't much how the method compares to full supervision or to other methods trained with the same amount of sparse labels. We only get to know the hyperparameters influence the method
>      - We agree that comparisons between these should be included and we have added a comparison to 3D U-Net on the sparse data. See the comment to all reviewers and the appendix for more detailed metrics. Full supervision performance using the multi-planar method on the cardiac dataset is outlined in the multi-planar U-Net paper (Perslev, et al.). We see full supervision dice performance on the cardiac dataset of approximately 0.89 and our method (when $n_{annot}=16$, and $n_{train}=512$) we achieve approximately 0.83. Additionally, on the mitochondria dataset we achieve a dice score of 0.9245 which is comparable to the 0.9288 achieved with full supervision (Lucchi et al).
> - To use the method in practice, the plane needs to be selected first, and a rater needs to annotate those planes subsequently. The process is not ideal: it cannot just leverage already existing sparsely labeled data.
>      - We can see how this may be unclear, so to clarify we can use already existing sparely labeled data. If we already have a sparsely-annotated volume, it is very easy to generate training planes by simply slicing through this sparsely annotated volume along random angles. Our method only generates the annotation planes in this manner to simulate a dataset that has been sparsely annotated along planes. One technique we envision is in an active learning environment which gives the user randomly oriented slices and asks them to annotate them thus generating a dataset in a similar manner.
>
> Responses to "Detailed comments"
> - The authors state " As a result, the multi-planar method is better at learning 3D structures in the data regardless of their local orientations." There is no results to support this in the paper. And intuitively it is strange to expect a 2D model to learn 3D shapes.
>      - We agree that the statement you mention is a bit unclear. We have corrected it to "As a result, the multi-planar method is better than a standard 2D U-Net at learning 3D structures in the data regardless of their local orientations." As previously stated, if 3D images can be represented as 2D data (as in 3D reconstruction), we expect that it is possible for a model to learn 3D structures as a 2D representation.
>
> Responses to "Questions To Address In The Rebuttal"
> - As suggested we have applied the technique to another dataset. Please see the appendix and the comment to all reviewers for a detailed explanation of the results. In summary, when applied to a cardiac MRI dataset we achieve similar convergence plots showing that we can get close to the performance of 16 manually annotated slices using only 4 annotated slices.
> - We have also compared to the 3D U-Net further validation. Please see the appendix and the comment to all reviewers for more information. In summary, the 3D U-Net performs similarly to our method with the downside of a longer training time.
>
> Thank you for the constructive review.

---

> > ### Comment · AnonReviewer6 · 2021-03-22
> > **Response of AnonReviewer6**
> >
> > Thank you for your efforts in answering my comments and those of the other reviewers.
> > * In essence, a model which only accept 2D input cannot learn 3D shapes. For example, how do make the difference between an ellipsoid and a cylinder only using 2D planes? Any section of each of the volumes are ellipses.
> > * Nice that you added another dataset, it makes the paper much stronger!
> > * Without statistical tests you cannot claim anything. I think it is essential. You could at least report confidence intervals for the reader to evaluate significance.
> > * It seems that a lot of the methodological novelty of the work was already presented by Perslev et al. 2019. But the article is written as this being a new method, which is why all the reviewers asked to see thorough comparison with fully supervised 3D U-Nets. I think it should be clarified in the abstract and introduction that this paper only evaluated the influence of hyperparameters of the previously published method (Perslev et al. 2019).
> > * I think the authors should mention more clearly in the paper--in the abstract and introduction--that the method is designed to be an active learning method.

---

> > > ### Author Response · Authors · 2021-03-23
> > > **Comment for AnnonReviewer6**
> > >
> > > Thank you for the insightful questions and thoughts.
> > >
> > > - In essence, a model which only accept 2D input cannot learn 3D shapes. For example, how do make the difference between an ellipsoid and a cylinder only using 2D planes? Any section of each of the volumes are ellipses.
> > >    - Recognition rates seem to be similar in both the 3D U-Net and the multi-planar U-Net. This indicates that either it is less important in practice or that the combination of different views during prediction captures the essential 3D information. In the experiments we have performed in the paper, we predict 3 randomly oriented 2D slices and average the results. The distribution of apparent ellipses in different 2D slices of an ellipsoid and a cylinder is quite different. For cylinders it is unbounded, and hence, an estimator will tend to estimate shapes indistinguishable from very large ellipsoids, possibly converging to shapes that are infinitely long. Any method using a finite field of view (including 3D U-Nets) also cannot distinguish cylinders from very elongated ellipsoids. We find this an extremely interesting case to consider in future research and we thank the reviewer for this inspiring question.
> > >
> > > - Without statistical tests you cannot claim anything. I think it is essential. You could at least report confidence intervals for the reader to evaluate significance.
> > >    - We agree that statistical tests are a useful way to summarize the comparison of distributions. In the article we have supplied the sample size, mean and standard deviation of the Dice scores. The p-values from the two-sided t-test can be evaluated from these. There are many combinations in our data that can be analyzed. For example, (4,128) vs (16,512) has a t-value of -5.7708, Welch-Satterthwaite estimated degrees of freedom of 15.1 and an ensuing p-value of 0.00003694. Hence we can reject that the two mean Dice scores are equal with a confidence of 0.05. However, this is not surprising and hardly needs a statistical test, since a normal distribution 0.88881+-0.0200 clearly has a very small overlap with 0.9245+-0.0088. We have also tested (4,128) versus (8,256) and (12,384) and (16,512) respectively and only for the combination (4,128) versus (8,256) can the null-hypothesis not be rejected with confidence of 0.05. Finally, we tested (4,0) vs (4,128) and (8,0) versus (8,256) etc. In which case, all null-hypotheses could be rejected with 0.05 confidence. Since we have not had time to run the 3D U-Net many times, we have not yet performed t-tests comparing the multi-planar U-Net with the 3D U-Net. However, for this paper, this of less importance since this paper is the study of the convergence of the performance of the multi-planar U-Net with regards to sparsity. If the paper is accepted, we will add these t-test results in the appendix.
> > >
> > > - It seems that a lot of the methodological novelty of the work was already presented by Perslev et al. 2019. But the article is written as this being a new method, which is why all the reviewers asked to see thorough comparison with fully supervised 3D U-Nets. I think it should be clarified in the abstract and introduction that this paper only evaluated the influence of hyperparameters of the previously published method (Perslev et al. 2019).
> > >    - Thank you for highlighting this and if accepted, we will improve the sentence in the introduction: “In this paper, we investigate the multi-planar U-Net’s (Perslev et al., 2019) abilityto learn from sparse annotations.” by including an explicit mentioning of the hyper-parameters as suggested by the reviewer.
> > >
> > > - I think the authors should mention more clearly in the paper--in the abstract and introduction--that the method is designed to be an active learning method.
> > >    - Active learning is one possible use for this, however, again we emphasize that this paper is a study on the convergence of performance with regards to sparsity, as emphasized by the title of the paper.

---

### Official Review · AnonReviewer4 · 2021-03-10

**Confidence:** 3
**Preliminary Rating:** 3

**Summary:**

This submission explores the required annotations for a 3D image segmentation using a multi-planar U-Net. The proposed method was evaluated on the Mitochondrial binary segmentation problem. The paper claims that less than 30% of the annotations are required to achieve comparable performance with multi-planar U-Net.


**Strengths:**

The submission targets an important issue in medical image segmentation. It provides a potential solution to address the annotation problem. The paper demonstrates a promising approach to greatly reduce the requirement of segmentation annotations.

**Weaknesses:**

The writing of this paper could be improved to increase its readability. Comparison with other sparse annotation methods is desired to make the paper solid. Working on more than one dataset will be more persuasive to demonstrate the effectiveness of this proposed mehtod.

**Deanonymize Review:**

no

**Justification Of The Preliminary Rating:**

Overall, the proposed idea in this paper is interesting and promising. However, it'll be better with more experiments on more than one dataset. The readability of this paper has room to further improve. I actually hold a borderline opinion on this paper. Unfortunately, there is no such a choice.

**Paper Type:**

methodological development

**Special Issue:**

no

---

> ### Author Response · Authors · 2021-03-17
> **Comment for AnonReviewer4**
>
> We would like to thank you for the feedback. We have taken it into consideration and made some necessary changes.
>
> Please read the comment to all reviewers first for some important changes/clarifications.
>
> - We agree that testing our method on only one dataset does not provide a solid standing for our method. As suggested we have applied our method to another dataset. The secondary dataset is the cardiac dataset from the 2018 Medical Decathlon challenge from http://medicaldecathlon.com/ (Simpson et al, 2019). See the comment to all reviewers for a summary of the results and more detailed metrics in the Appendix of the revised paper. In summary, the method improves the dice score in a similar manner as we found on the mitochondria dataset.
> - We also agree that a comparison with 3D U-Net would also be useful. We have therefore included this in the appendix and outlined the results in the comment to all reviewers. In summary, the 3D U-Net performs similarly to the multi-planar version with the downside of longer training times when using the 3D version.

---

### Official Review · AnonReviewer5 · 2021-03-10

**Confidence:** 4
**Preliminary Rating:** 3
**Recommendation:** Poster
**Final Rating:** 3

**Summary:**

The authors propose fusing a multi-planar data augmentation scheme with sparsely labelled annotation masks for binary segmentation of mitochondria in volumetric electron microscopy images. The ground truth segmentation masks are obtained by a two-step approach. First, randomly oriented 2D slices are annotated by an expert. Second, other randomly oriented slices are selected for training, meaning that annotated pixels are only available on intersection lines with the annotation slices.
The presented work should show how multi-planar data augmentation can improve binary segmentation results, when dealing with (artificial) data scarcity.
In general, I welcome investigating approaches for improving segmentation performance on small datasets. While the paper overall is presenting relevant information to the community, is well structured and easy to follow, there are some issues which need to be considered in a major revision.

**Strengths:**

In my opinion, developing approaches for improving neural network performances on limited data is an important field of study, especially in the medical domain. The method proposed by the authors could define a point from where to start more detailed investigations and exploration of explicit applications. The content is easy to follow and well structured. Almost all relevant information for understanding the details of the proposed approach is given in the text.

**Weaknesses:**

A single metric is not enough for assessing segmentation algorithms in a meaningful manner. The Dice score doesn't provide information about edge alignment which is crucial in many medical imaging applications. I strongly suggest augmenting the set of evaluation metrics by at least a single measure of edge alignment like the average Hausdorff distance or the average symmetric surface distance. See [Taha et al. Metrics for evaluating 3D medical image segmentation: analysis, selection, and tool. BMC Medical Imaging. 2015] for further segmentation metrics.


The authors justify the usage of a 2D U-Net with the large computational requirements of 3D U-Nets. This seems to be correct intuitively, but also depends on the size of input volume crops which could be chosen such that the computational cost would be roughly equivalent to the 2D case. It then remains the question if the 3D U-Net outperforms the 2D U-Net. Therefore, a comparison with a corresponding 3D-U-Net is strongly recommended from my side to show the usefulness of the proposed method. Additionally, further information should be given on VRAM load, training time as well as inference time for both 2D and 3D U-Net. Especially, inference time would be interesting due to the exhaustive testing scheme of the 2D case.

The authors should try to further bring their findings into a clinical context and discuss possible application scenarios. Are there cases, where the acquisition of sparse labels in 3D volumes would justify the diminished performance of the resulting CNNs? This also relates to the next point.

I recommend comparing the results by the proposed method to results of previous works. At least to the original paper cited by the authors [Lucchi et al. Learning for Structured Prediction Using Approximate Subgradient Descent with Working Sets. IEEE Conference on Computer Vision and Pattern Recognition. 2013]. Here, a Dice score of 92,9 % (Jaccard: 86,7 %) is achieved. Of course, not under the constraint of sparse labels. Nevertheless, this raises the question if the presented results are sufficient for clinical applications. Additionally, the results of a 2D and 3D U-Net trained with all labels should be added as a reference.

**Deanonymize Review:**

no

**Detailed Comments:**

I strongly recommend not calling the U-Net "multi-planar" because it's not a property of the network. "Multi-planer" is rather a term describing the data augmentation scheme. For clarification, calling a network "random rotation" U-Net doesn't seem to be very favorable when random rotations are performed for data augmentation.

I suggest adding exemplary segmentation images to the appendix to give the reader a visual impression of the results. Especially, when compared to the case without sparse annotations.

In the abstract, the authors state that multi-planar data augmentation for 2D U-Nets reduces the amount of needed training data compared to the 3D case. I would like the authors to explain this statement.

In section 3, the authors say that the dataset consists of two volumes. One for training and another for testing. However, section 4 talks about multiple training volumes. Are the "large" volumes divided into sub-volumes?

I guess there is a typo in the given image volume size of 165 x 2048 x 1536. It should rather be 1065 x 2048 x 1536.

Axis labels in Figure 3 would be helpful.

**Final Rating Justification:**

The authors addressed most of my concerns and delivered a paper that is suitable for sharing with the scientific community. The MIDL conference provides a good environment to discuss this approach further.

**Justification Of The Preliminary Rating:**

In general, the paper is well written and provides interesting content for the scientific community. Nevertheless, the findings appear somewhat isolated and are not compared to alternative methods like ordinary 3D U-Nets. Furthermore, the paper lacks a discussion of the clinical applicability regarding the trade-off between the amount of annotations and the resulting segmentation performance. However, in my opinion, the paper suits the scope of MIDL and should be presented.

**Paper Type:**

methodological development

**Questions To Address In The Rebuttal:**

Comparison with a 3D U-Net.

Embed findings into clinical context and discuss possible applications where the trade-off between sparse annoations and dimished performance is acceptable.

Adding a segmentation metric of edge alignment.

**Special Issue:**

no

---

> ### Author Response · Authors · 2021-03-17
> **Comment for AnonReviewer5**
>
> We would like to thank the reviewer for the detailed feedback and will take it into account in future revisions.
>
> Please read the comment we have made to all reviewers first as it contains some important information.
>
> Responses to "Questions to address in rebuttal":
> - 3D UNet - We agree that a comparison to 3D U-Net would be helpful. As such, we have added a comparison to the 3D U-Net to the Appendix and is also described in the comment to all reviewers. The basic overview is that the multi-planar version and the 3D U-Net have similar performance, however, the longer training time in the 3D model is detrimental.
> - Clinical context - The lower performance of our method has been corrected with a bug fix described in the comment to all reviewers, however, we believe this is still a good question to address. Our paper is a study on the convergence of different levels of sparsity in the context of the multi-planar U-Ńet. The convergence point (when trained on fully supervised data) has been indicated in the original multi-planar article (Perslev, et al). In a clinical use case, we would always have to make a trade-off between the amount of time spent on annotation and the quality of the result. An ideal use case might be in an active learning environment where the benefits of faster training are more prominent and the user could annotate until results are satisfactory.
> - Edge metrics - We see the importance in looking at an edge metric in addition to dice. As such we have re-run the experiments and recorded the average surface distance for the various predictions. Average surface distance from ground truth to predicted masks is less than 1 in nearly all cases (except ($n_{annot}=4$, $n_{train}=0$) and ($n_{annot}=8$, $n_{train}=0$) with 5.4015 and 1.0518 respectively) with the best result of 0.5245 occurring with $n_{annot}=16$, and $n_{train}=512$. The average surface distance as computed from prediction to ground truth are much higher (30.5104 with $n_{annot}=4$, and $n_{train}=0$ and 13.9740 with $n_{annot}=16$, and $n_{train}=512$). Upon further inspection this is highly dependent on which 3 views are chosen during prediction that sometimes produce small rough outlier blobs with high surface to volume ratios. This can be seen in some of the predicted masks that we have added to the Appendix. Many of these "outlier" blobs could relatively easily be caught using post-processing via morphological filtering and/or small object removal. We did not have time investigate the improvement that could be gained via post-processing in detail as this paper is intended to be an analysis of our technique alone. However a few quick runs suggest that doing connected components and saving the largest objects seems to improve this fairly significantly (by half in some cases). Both of these average surface distance results seem to follow the same trend as the dice score, improving as both $n_{annot}$ and $n_{train}$ increase. Surface distance metrics have also been added to the Appendix.
>
>
> Other clarifications:
> - Our statement that this method reduces the number of annotations needed versus the 3D case is made under the assumption that the 3D case is trained using fully segmented volumes (not sparse volumes). Our comparison between the 3D-U-Net on the sparse samples also suggests that this method also seems to perform similarly to the 3D U-Net while having shorter training time.
> - You are correct that there is a typo in the volume dimensions in the paper, however not as you suggest. The dimensions should actually be 165 x 768 x 1024. This is because the creators of the dataset only annotated the first 165 slices of the 1065 x 2048 x 1536 volume and then split it into training and testing volumes. The original training and testing volumes are in fact 165 x 768 x 1024. The training volume has been split into 4 sub-volumes of dimensions 165 x 448 x 448. The 12 experimental runs were created via train-validation pairs from these 4 volumes (each model was trained using a single training volume and a single validation volume). These have been corrected and clarified in the revised paper.
> - Example predictions have been added to the appendix.
> - Axis labels have also been added to all figures for clarification.
> - A comparison to the original paper [Lucchi et al. Learning for Structured Prediction Using Approximate Subgradient Descent with Working Sets. IEEE Conference on Computer Vision and Pattern Recognition. 2013] has been mentioned in the updated version.
> - Computational efficiency - It takes approximately 2 hours and 45 minutes to train the 3D version and approximately 1 hour and 5 minutes to train the 2D version on a Titan RTX. Predicting a single 165x165x165 volume takes approximately 45 seconds using the multi-planar version and 25 seconds using the 3D version.

---

> > ### Comment · AnonReviewer5 · 2021-03-18
> > **Integration of Additional Results**
> >
> > It is good to see that the authors adressed nearly all of my comments. Adding results from a 3D U-Net as well as further segmentation metrics increased the paper's informative value. Apart from that, a final evaluation of which of the two methods leads to better results is not possible due to lack of statistical significance. Furthermore, the additional results and metrics are not really discussed and integrated into the conclusion. They appear in the appendix somewhat isolated. I would suggest to remove the whole part of fitting the Dice score, which doesn't essentially give additional insight (saturation is expected and can also be derived by looking at the numbers), and instead augment discussion and conclusion regarding the added results and segmentation metrics.
> >
> > I would like the authors to comment on the first statement I made in the "Detailed Comments" section. This is not an important point, but I am very interested in the authors' opinion.

---

> > > ### Author Response · Authors · 2021-03-19
> > > **Comment for Integretation of Additional Results**
> > >
> > > We agree that mentioning and discussing the results of the 3D U-Net in the paper would be nice. As such we made one last modification yesterday to include a short discussion of the results in a paragraph in the results section. However, they are not discussed in detail in the conclusion. The reason for this is both due to space limitations and the fact that the primary purpose of the paper is to analyze how the sparsity affects the performance of the multi-planar U-Net and the comparison to the 3D U-Net is supplementary. Results from the original multi-planar paper (Perslev et al, 2019) already show the effectiveness of the multi-planar technique on 3D data as does their results in the 2018 Medical Decathlon challenge at http://medicaldecathlon.com/results.html (team name is CerebriuDIKU).
> > >
> > > Regarding the naming of "multi-planar U-Net", we agree that it may not be ideal to name the technique in such a way as it is indeed not a property of the U-Net itself, however the naming is done in this way to follow the convention defined in the original multi-planar paper (Perslev et al, 2019). A secondary reason for this is that writing out "U-Net with multi-planar data augmentation" is longer and less concise than just using "multi-planar U-Net", especially with a limited number of pages.

---

### Author Response · Authors · 2021-03-17
**Comment for all reviewers - Please read this first - Part 1**

We would like to thank the reviewers for providing valuable feedback with regards to our paper.

In the following, we make general clarifications and describe all major changes made to the manuscript since review. This is a general comment to all reviewers. We address specific feedback from each reviewer in individual comments as well.

We have made the following three major changes to the manuscript:

1.) Correction of evaluation code bug:
Upon further evaluation of our code we realized we had mistakenly used the wrong prediction function to compute the results. During prediction along each view, the volume is rotated in full. The function we used in the results displayed in the original version of the paper does not pad each volume prior to rotation. This has resulted in a large number of pixels located at the corners of the volume being cut off causing approximately 35\%-40\% of pixels in the volumes to incorrectly return 0 during prediction. Upon correction of this error, the overall conclusion and discussion points remain unchanged, however the dice scores have increased significantly (dice score improved from the 0.7-0.76 range to the 0.89-0.92 range). See the revised paper for updated results.

2.) Comparison to 3D U-net:
We have added a comparison between the 3D U-Net and our proposed method. We used the 3D U-Net as defined in the original 3D U-Net paper (Cicek et al) with approx 19 million parameters. We used an input size of 64 x 64 x 64 and trained for 22400 batch updates using a batch size of 1 with weights producing the best validation score saved for further analysis on the held-out test set. The input size was chosen such that the number of voxels viewed during each weight update is similar to that of the multi-planar version (Multi-planar: 262144 and 3D: 262144) to ensure a similar amount of information is learned at each weight update. The models were trained on the same data seen by the multi-planar version (i.e. the annotated voxels are the ones that appear in the intersection of the $n_{annot}$ voxels and the $n_{train}$ voxels when $n_{train} > 0$ and the $n_{annot}$ voxels alone when $n_{train} = 0$). Data augmentations are the same as those in the multi-planar version.

|            | n_train=0 | n_train=128 | n_train=256 | n_train=384 | n_train=512 |
|------------|-----------|-------------|-------------|-------------|-------------|
| n_annot=4  | 0.8419    | 0.8251      | 0.8836      | 0.9049      | 0.8299      |
| n_annot=8  | 0.8519    | 0.8730      | 0.8424      | 0.8737      | 0.8975      |
| n_annot=12 | 0.8719    | 0.8028      | 0.8586      | 0.9062      | 0.9180      |
| n_annot=16 | 0.8783    | 0.8741      | 0.9209      | 0.9178      | 0.8466      |

The best scores in each row of the 3D U-Net plot above are comparable to the best score in each row of the multi-planar version. The high variability is due to only having time to compute results for a single run. The training time for the 3D model takes approximately 2 hours and 45 minutes, while the multi-planar version takes only 1 hour and 5 minutes.

3.) Evaluation of an additional dataset:
We have trained and evaluated our proposed method on an additional dataset. Specifically, we considered the cardiac mono-modal MRI dataset from the 2018 Medical Decathlon Challenge (Simpson et al, 2019). As we did not have access to the ground truths for the test set, we divided the 20 training volumes into train, validation and test sets. Four volumes were set aside and used as the held-out test set. The remaining 16 volumes were divided into 12 training volumes and 4 validation volumes. This is repeated 12 times with different train-validation splits.

|            | n_train=0 | n_train=128 | n_train=256 | n_train=384 | n_train=512 |
|------------|-----------|-------------|-------------|-------------|-------------|
| n_annot=4  | 0.5564    | 0.7500      | 0.7371      | 0.7414      | 0.7410      |
| n_annot=8  | 0.6401    | 0.7758      | 0.7762      | 0.7872      | 0.7969      |
| n_annot=12 | 0.7075    | 0.8071      | 0.7955      | 0.7791      | 0.7833      |
| n_annot=16 | 0.7510    | 0.8121      | 0.8211      | 0.7989      | 0.8314      |

With $n_{annot} = 16$ and $n_{train} = 512$ we achieve a dice score of 0.83 compared to a fully supervised multi-planar model which achieves approximately 0.89 (see Perslev et al, 2019). Overall, the trend follows a similar pattern to results on the mitochondria dataset with $n_{annot} = 4$ and $n_{train} = 512$ achieving close to the performance of $n_{annot} = 12$ and $n_{train} = 0$ (dice scores of 0.7410 and 0.7510 respectively) while beating out $n_{annot} = 12$ and $n_{train} = 0$ (dice score 0.7075) fairly significantly. This further solidifies the argument that our method significantly reduces the amount of annotations needed. This secondary dataset was chosen as it has significant structural differences along different viewing angles compared to the mitochondria dataset.

---

### Author Response · Authors · 2021-03-18
**Comment for all reviewers - Please read this first - Part 2**

We have discovered as the reviewers have predicted that there are some interesting relations between the multi-planar U-Net and the 3D U-Net. Due to lack of space, we have been forced to put the 3D U-Net results in the appendix. Due to lack of time we have not been able to fully investigate these relations, however, our initial interpretations have been added to the main paper in a paragraph in the results section and refer to the results in the appendix. They are that the 3D U-Net also performs well on these very sparse datasets, however, it seems that the variability of the dice score is greater in the 3D U-Net than the multi-planar U-Net. The 3D U-Net is also slower to train, however we have not had time to optimize the training. Nevertheless, we underline that the purpose of this paper is to study the convergence rates of the multi-planar U-Net with respect to the size of the sparse training set, and the comparison to 3D U-Net however important is secondary.

---

### Meta-Review · Area_Chair1 · 2021-03-30

**Recommendation:** Reject

**Metareview:**

All reviewers agree that the proposed work is of great importance to the problem of image segmentation, particularly under the condition where limited number of annotations are available. The proposed work has decent quality of experimental design. However, the reviewers also have raised several concerns of the paper, majorly in the limited novelty of the methodological development (as the core idea was published in previous conference of MICCAI). In addition, while the authors added experimental results validated on more datasets, the small number of training images (12 cardiac MRIs) makes the results of such deep learning-based approach less convincing. Overall, based on the final ratings of all reviewers’, the current submission will need further improvement to publish at MIDL.

**Paper Type:**

methodological development

---

### Decision · Program_Chairs · 2021-03-31

Reject